# High-Fat Diets Modify the Proteolytic Activities of Dipeptidyl-Peptidase IV and the Regulatory Enzymes of the Renin–Angiotensin System in Cardiovascular Tissues of Adult Wistar Rats

**DOI:** 10.3390/biomedicines9091149

**Published:** 2021-09-03

**Authors:** Germán Domínguez-Vías, Ana Belén Segarra, Manuel Ramírez-Sánchez, Isabel Prieto

**Affiliations:** 1Unit of Physiology, Department of Health Sciences, University of Jaén, Las Lagunillas, 23071 Jaén, Spain; asegarra@ujaen.es (A.B.S.); msanchez@ujaen.es (M.R.-S.); 2Department of Physiology, Faculty of Health Sciences, Ceuta, University of Granada, 18071 Granada, Spain

**Keywords:** aminopeptidase activities, angiotensinases, dipeptidyl peptidase IV, high-fat diet, virgin olive oil

## Abstract

(1) Background: The replacement of diets high in saturated fat (SAFA) with monounsaturated fatty acids (MUFA) is associated with better cardiovascular function and is related to the modulation of the activity of the local renin–angiotensin system (RAS) and the collagenase activity of dipeptidyl peptidase IV (DPP-IV). The objective of the work was to verify the capacity of different types of dietary fat on the regulatory activities of RAS and DPP-IV. (2) Methods: Male Wistar rats were fed for 24 weeks with three different diets: the standard diet (S), the standard diet supplemented with virgin olive oil (20%) (VOO), or with butter (20%) plus cholesterol (0.1%) (Bch). The proteolytic activities were determined by fluorometric methods in the soluble (sol) and membrane-bound (mb) fractions of the left ventricle and atrium, aorta, and plasma samples. (3) Results: With the VOO diet, angiotensinase values were significantly lower than with the Bch diet in the aorta (GluAP and ArgAP (mb)), ventricle (ArgAP (mb)) and atrium (CysAP (sol)). Significant decreases in DPP-IV (mb) activity occurred with the Bch diet in the atrium and aorta. The VOO diet significantly reduced the activity of the cardiac damage marker LeuAP (mb) in the ventricle and aorta, except for LeuAP (sol) in the ventricle, which was reduced with the Bch diet. (4) Conclusions: The introduction into the diet of a source rich in MUFA would have a beneficial cardiovascular effect on RAS homeostasis and cardiovascular functional stability.

## 1. Introduction

The high consumption of a high-fat diet (HFD) has been associated with most of the epidemiological evidence during the last decades, especially the sources of saturated fatty acids (SAFA) and cholesterol have been associated with an increased risk of suffering a series of medical conditions such as obesity, diabetes, metabolic syndrome (MetS), and cardiovascular diseases (CVD); however, these results do not extend to other types of lipid sources [1]. HFD is a cardiac stressor that alters the expression of specific cardiac factors due to glycolipotoxicity [2] and modifies cardiac structure and function. However, trials involving the Mediterranean diet indicate that monounsaturated fatty acids (MUFAs) are more effective than low-fat and low-cholesterol diets in preventing cardiovascular mortality and coronary artery disease [3]. The reduction in SAFA in substitution by MUFA attenuates the increase in blood pressure (BP) [4]. These alterations, depending on the degree of fatty acid saturation, are related to changes in the systemic or local renin–angiotensin systems (RAS) [5,6,7,8,9,10]. The activation of the RAS participates in the development of MetS, heart failure [11], and pathophysiology of hypertension [4,7,12,13,14,15,16,17,18,19,20,21,22,23,24,25]. Within these local RAS, the type of fatty acids consumed with the diet allows modifying various enzymes of the aminopeptidase (AP) family, also called angiotensinases, responsible for metabolizing angiotensin peptides (Ang) [5,9,26]. The APs are relevant in the control of BP and cardiac function [6,27], participating in the regulation of systemic and local RAS, but also as predictive biomarkers of functionality in different organs, including the heart [28,29,30,31,32,33,34,35]. It is suggested that several of the activities of APs involved in the metabolism of angiotensin in the cardiovascular tissues of the rat can be modified according to the amount and type of fat in the diet [6]. Among them, the activity of aspartyl-AP (AspAP; EC 3.4.11.21) is responsible for the metabolism of Ang I to Ang 2–10; glutamyl-AP (GluAP or aminopeptidase A, APA; EC 3.4.11.7) metabolizes Ang II into Ang III; alanyl-AP (AlaAP or aminopeptidase M, APM; EC 3.4.11.2) and/or arginyl-AP (ArgAP or aminopeptidase B, APB; EC 3.4.11.6) are responsible for the metabolism of Ang III to Ang IV and Ang 4–8; and insulin-regulated AP (IRAP; 3.4.11.3)—also called AP-placental leucyl (LAP), cystinyl-AP (CysAP), oxytokinase, or vasopressinase—was identified as the binding site for the Ang IV receptor (AT4) [12,29,36,37,38,39].

Ang II and aldosterone are produced de novo within cardiac tissue [40,41]. RAS activation by diets high in SAFA induces CVD through the manifestation of oxidative stress, inflammation, interstitial fibrosis at sites of vascular injury, ultrastructural abnormalities, and diastolic dysfunction [42,43]. Hemodynamic factors regulate the growth of cardiomyocytes, whereas reactive fibrosis is independent of blood pressure [40]. Under physiological conditions the fibroblasts of the cardiac interstitium explain the gradual turnover of collagen for a correct remodeling of the heart, but an alteration of the RAS would modify the balance between the restructuring behavior [43,44] and collagen degradation [45,46]. Certain APs with broad specificity affect collagen production in cardiac fibroblasts [47]. Dipeptidyl-peptidase IV (DPP-IV; EC 3.4.14.5), also known as collagenase-like peptidase, is a ubiquitously expressed serine-type aminopeptidase enzyme, with subsequent cleavage at the N-terminal amino acids proline and alanine (X-Pro and X-Ala), and also metabolizes the insulinotropic hormone glucagon-like peptide-1 (GLP-1) [48,49,50]. DPP-IV manifests itself on the surface of vascular endothelial cells and in renovascular organs, suggesting a role for DPP-IV in renal and cardiovascular function [51,52], in addition to being associated with ECV [53,54]. The cardiac fibroblast migration and expressions of genes related to fibrosis, such as collagen-I and transforming growth factor-β1 (TGF-β1), are suppressed with a DPP-IV inhibitor [42,55], which, in turn, inhibits MR-dependent oxidative stress and collagen induction mediated by aldosterone. 

Thyrotropin-releasing hormone (TRH) has also been identified in the heart and aorta [56], where it can modulate cardiovascular function, either autocrine or paracrine. The enzyme pyroglutamyl-AP (pGluAP; EC 3.4.11.8) is known for its thyrotropin-releasing hormone (TRH) degradation activity [18]. Previous results suggest that the type of fat used in the diet can influence the local activity of pGluAP and modify its biological functions in energy metabolism [4,9,57]. However, its effect on cardiovascular tissues is not entirely clear.

Along with angiotensinases, there are other APs such as leucyl-AP (LeuAP; EC 3.4.11.1) and gamma-glutamyl transferase (GGT; EC 2.3.2.2) involved in the control of BP that act as functional markers of cardiovascular risk [7,9,58] and that are associated with different cardiovascular and kidney diseases [34,54]. GGT is an enzyme of glutathione and cysteine (Cys) metabolism with pro-oxidant activity and a modulating influence on endothelial dysfunction [59]. Elevation of GGT levels is associated with an increased risk of cardiovascular and metabolic diseases [59,60,61,62]. GGT is associated with an increase in mortality from chronic heart disease events such as hypertension and congestive heart failure [28,31,63,64].

The main objectives of the current research were to understand the role of two types of diets with higher MUFA or SAFA fat content on the angiotensinase activities that regulate the classic and non-classic RAS pathways, to identify the implications for DPP-IV collagenase activity, and to verify in each dietary group if there is an inter-relationship between DPP-IV activities and angiotensinases in cardiovascular tissues. Measuring the activities of GGT and LeuAP biomarkers and the degrading activity of TRH (pGluAP) would make it possible to know the functional and metabolic state of cardiovascular tissues. We hypothesized that the inclusion of MUFA in the diet could be a preventive dietary factor for reducing the activation of the RAS pathway through angiotensinases and maintaining the functional integrity of cardiovascular structures.

## 2. Materials and Methods

### 2.1. Animals and Diets

Harlan Interfauna Ibérica SA (Barcelona, Spain) supplied adult male Wistar rats for experimentation. The experimental procedures for the use and care of animals were carried out in accordance with Directive 2010/63/EU of the Council of the European Communities and Spanish Regulation RD 53/2013, being previously approved by the Institutional Committee for Animal Use and Care of the University of Jaén with project code number PIUJA_2005_acción 14 (1 January 2006). The rats had the experimental diets and water ad libitum for 24 weeks and were kept under a controlled environment of temperature (20–25 °C) and humidity (50 ± 5%) in a 12 h:12 h light/dark cycle. At the start of the research, the animals were 6 months old and had an average body weight of ± 495 g. The rats were randomly assigned into three experimental groups: (1) A standard diet (S, *n* = 6), where the rats were fed with commercial food for laboratory rodents (Panlab, Barcelona, Spain) whose nutritional composition was 16.5% protein, 3% total fat, 60% carbohydrates (nitrogen-free extract (NFE)), 5% minerals, and 4% fiber. The other two diets were high in fat (HFD), but with different fatty acid composition. (2) A group of rats was fed diet S supplemented with 20% virgin olive oil (Cooperativa de los Villares, Jaén, Spain) (VOO, *n* = 5), composed of a total content (%) of 75.5% ω-9-monounsaturated fatty acid (oleic acid, C18:1), 11.5% saturated fatty acid (palmitic acid, C16:0), and 7.5% ω-6-polyunsaturated fatty acid (linoleic acid, C18:2). (3) The second group of rats was fed diet S supplemented with 20% butter (Hacendado, Valencia, Spain) plus cholesterol (0.1%) (Bch, *n* = 5), composed of a total content (%) of 29% monounsaturated fatty acid (C18:1), 62% saturated fatty acid (C16:0 and C18:0), 4% polyunsaturated fatty acid (C16:0), and short and medium chain fatty acids (C4–C14). The Bch diet was supplemented with cholesterol (0.1%) in order to reach the average cholesterol content of the Western diet. The HFD diets were isocaloric among themselves (VOO diet, 1848 KJ/100 g; diet Bch, 1827 KJ/100 g) and hypercaloric compared with the S diet (1392 KJ/100 g).

At the end of the experimental period, the animals were perfused with a saline solution (0.9% NaCl) through the left heart ventricle under Equithensin anesthesia (2 mL/kg of body weight). A blood sample dissolved in heparin was previously extracted and centrifuged for 10 min at 2000× *g* to obtain plasma. The heart and aorta were dissected, immediately deposited in liquid nitrogen to separate the samples into different parts as previously described [21], and frozen at −80 °C until use. The atrium and left ventricle were dissected from the heart. The aorta was obtained by cutting from the aortic arch to its abdominal portion.

### 2.2. Assay of Aminopeptidase Activities

Tissue samples were homogenized in 10 volumes of 10 mM Tris-HCl buffer (pH 7.4) and ultracentrifuged at 100,000× *g* for 30 min at 4 °C. Analysis of enzyme activities and protein content of the fraction soluble were performed with the supernatant and analyzed in triplicate. The pellets were rehomogenized in 10 mM Tris-HCl buffer (pH 7.4) with 1% Triton-X-100 to solubilize the membrane proteins and ultracentrifuged at 100,000× *g* for 30 min at 4 °C. The supernatants were kept for at least 4 h at 4 °C and shaken with SM-2 biobeads (100 mg/mL) (Bio-Rad, Richmond, VA, USA) to remove the detergent. Analyses of enzyme activities and protein content of the membrane-bound fraction were conducted with the resulting samples, analyzed also in triplicate. 

The enzymatic activities of the soluble (sol) and membrane-bound (mb) fractions of AlaAP, ArgAP, AspAP, CysAP/IRAP, DPP-IV, GGT, GluAP, LeuAP, and pGluAP were measured by a fluorometric analysis using as aminoacyl-β-naphthylamides (aa-β-NA) substrates: L-Ala-β-NA, L-Arg-β-NA, L-Asp-β-NA, L-Cys-β-NA, L-Gly-Pro-β-NA, ɣ-Glu-β-NA, L-Glu-β-NA, L-Leu-β-NA, and L-pGlu-β-NA, respectively, according to the methods of different authors [65,66,67] modified by Prieto and Ramírez [15,68]. An amount of 10 µL/well of the supernatant sample was pipetted in 96-well black plates and was incubated for 30 min at 37 °C in 100 µL of substrate solutions. The enzymatic reactions were then stopped by adding 100 µL of 0.1 M acetate buffer (pH 4.2). The β-NA released as a result of the enzymatic activity was quantified fluorometrically at an emission of 412 nm with an excitation of 345 nm. The activities of each tissue fraction were expressed as pmoles of L-Ala-β-NA, L-Arg-β-NA, L-Asp-β-NA, L-Cys-β-NA, L-Gly-Pro-β-NA, ɣ-Glu-β-NA, L-Glu-β-NA, L-Leu-β-NA, and L-pGlu-β-NA hydrolyzed per minute and per mg of protein (pmol aa-β-NA/min/mg prot). For plasma, it was expressed as picomole per minute and per milliliter (pmol aa-β-NA/min/mL). All chemical products were supplied by Sigma-Aldrich (St. Louis, MO, USA).

### 2.3. Protein Measurement

For protein quantification, the Bradford method [69] was used, using as standard a bovine serum albumin (BSA; Sigma-Aldrich, St. Louis, MO, USA) standard line.

### 2.4. Statistical Analysis

Statistical analysis was performed using one-way ANOVA, followed by Tukey’s post hoc test for multiple comparisons. When the normality test failed, a Kruskal–Wallis unidirectional range analysis of variance was performed. Pearson’s correlation coefficient was used to establish the relationship between the APs activities of cardiovascular tissues and plasma. Significant differences were estimated with Sigmaplot v11.0 software (Systat Software, Inc., San José, CA, USA), and *p*-values less than 0.05 (*p* < 0.05) were considered statistically significant. All data are presented as mean ± standard error of the mean (SEM).

## 3. Results

### 3.1. Angiotensinase Activities

To know how the type of fat in the diet affected the angiotensinase activities that were derived from the classic RAS route, the activities that were studied are represented in the following scheme in green, following the order of action of each angiotensinase based on its precursors and metabolic products (Figure 1).

The values of all angiotensinase activities can be consulted in Appendix A. None of the angiotensinase activities were significant in plasma. No significant differences were observed in any of the tissues analyzed for AspAP and AlaAP activities. Furthermore, the AspAP activity in the soluble fractions of the atrium and aorta was undetectable. The VOO diet had significantly lower values than the Bch diet for the activities GluAP (mb) and ArgAP (mb) in the aorta (GluAP (mb), VOO: 2024.36 ± 420.28 vs. Bch: 3986.00 ± 582.40; ArgAP (mb), VOO: 1548.75 ± 129.86 vs. Bch: 2087.72 ± 139.98), ArgAP (mb) in the ventricle (VOO: 763.06 ± 15.88 vs. Bch: 1013.07 ± 56.57), and CysAP (sol) in the atrium (VOO: 195.50 ± 23.11 vs. Bch: 310.75 ± 21.51) (Figure 2A,C–E). However, in the ventricle, the ArgAP (sol) activity had a higher value with the VOO diet with respect to the Bch diet (VOO: 255.20 ± 52.95 vs. 127.80 ± 17.60; Figure 2B). Furthermore, the Bch diet significantly increased CysAP (sol) activity of the atrium compared with the S diet (S: 212.18 ± 32.74 vs. Bch: 310.75 ± 21.51; Figure 2E).

Appendix A shows the detection of grouped data correlations for GluAP (sol/mb) and ArgAP (mb) activities between the ventricle and aorta, for CysAP (mb) activity between the aorta and atrium, and for AlaAP (sol) between the plasma and ventricle. However, when the dietary groups were analyzed independently with a stratified analysis, it was observed that the association was lost.

### 3.2. Dipeptidyl Peptidase IV Activity

All values of DPP-IV activity can be found in Appendix A. Collagenase activity of DPP-IV showed significant changes with HFDs in the atrium and aorta (Figure 3). The atrium had higher values of DPP-IV activity (mb) in the VOO diet than in the Bch diet (VOO: 1743.86 ± 99.46 vs. Bch: 1222.50 ± 223.93; Figure 3A). On the other hand, in the aorta, the Bch diet reduced the DPP-IV activity (mb) with respect to the S diet (S: 3937.62 ± 478.42 vs. Bch: 2321.89 ± 224.54; Figure 3B). Despite a positive group correlation with all the data for DPP-IV (sol) activity between aorta and plasma, the stratified correlations did not show significance for independent groups (Appendix A). 

The DPP-IV activity showed significant grouped correlations of all the point data with the angiotensinase activities in the atrium, ventricle, and aorta (Table 1). However, a stratified analysis for dietary group comparisons showed that only the atrium and ventricle had strong correlations between activities (Table 1: intra-group correlations). DPP-IV and angiotensinase activities seem to be associated with diets, interestingly highlighting the association of the VOO diet in the soluble fractions of the atrium and ventricle. 

### 3.3. Leucyl Aminopeptidase, Gamma-Glutamyl Transferase, and Pyroglutamyl Aminopeptidase Activities

All values of LeuAP, GGT, and pGluAP activities can be found in Appendix A. Activity levels in the atrium were undetectable for LeuAP (sol/mb), GGT (sol/mb), and pGluAP (mb) activities. The GGT activity did not show that they were regulated by the different HFDs in the different samples examined. The VOO diet presented significantly lower LeuAP (mb) activity values than the Bch diet in the ventricle (VOO: 667.32 ± 23.08 vs. Bch: 924.99 ± 39.31; Figure 4B) and aorta (VOO: 1753.36 ± 213.53 vs. Bch: 2681.70 ± 111.30; Figure 4C). In turn, the Bch diet showed a significant increase in LeuAP (mb) activity compared with the S diet (Figure 4B, S: 636.06 ± 70.51 vs. Bch: 924.99 ± 39.31; Figure 4C, S: 2071.56 ± 100.69 vs. Bch: 2681.70 ± 111.30). On the contrary, LeuAP (sol) activity was significantly reduced with the Bch diet in the ventricle (S: 291.37 ± 33.40 vs. Bch: 162.87 ± 21.53; Figure 4A). With all the point data, a significantly positive grouped correlation was found between the LeuAP activity (mb) of the aorta and the ventricle, but a stratified analysis by dietary groups (Appendix A: intra-group correlations) showed that LeuAP was not associated with diet. Furthermore, both HFDs showed a reduction in pGluAP (mb) activity only in the atrium (S: 75.99 ± 1.82; VOO: 58.22 ± 4.92; Bch: 54.83 ± 6.00; Figure 4D).

## 4. Discussion

Adequate nutritional status is essential for maintaining cardiovascular structure and functionality in the face of metabolic challenges [2,70]. During the last decade, there has been intensive debate about the advice to reduce SAFA and increase MUFA or polyunsaturated (PUFA) to reduce the risk of CVD [71,72]. Virgin olive oil has a preventive effect on the development of atherosclerosis, indicating that endothelial damage triggered by oxidation can be diminished or reversed by the compounds of olive oil [73,74]. In recent work in our laboratory using the same HFDs (VOO and Bch), we confirmed the beneficial effects in animals fed the VOO diet vs. the pernicious results with the Bch diet. With the VOO diet, they do not increase their body weight, they do not increase the values of systolic blood pressure (SBP), and they maintain normal values of triglycerides and the fraction of VLDL cholesterol, reduce total cholesterol and its LDL fraction, and reduce oxidative stress and lipid peroxidation against the pernicious effects that the Bch diet manifests. [4,7,9]. Elevated plasma cholesterol levels are another important risk factor widely recognized for its relationship with angiotensin metabolism [4,5,7,10,12,72], with Mediterranean countries showing lower rates of heart disease than other countries due to the usual diet rich in olive oil [75]. 

HFDs alter the baroreceptor reflex [76], and the type of fatty acid that makes up the diet is capable of modulating the central [10,26,77] and local RAS regulatory APs [4,5,7,9,57]. Our results show that the VOO diet favored the stabilization of the RAS with respect to the Bch diet, by lowering the values of GluAP (mb) activities in the aorta, ArgAP (mb) in the ventricle and aorta, and CysAP (mb) in the atrium. In turn, it was observed in the ventricle that CysAP (mb) activity was significantly elevated with diet S in addition to VOO. These data may have repercussions with previously published data, where they identified that the local and serum mRNA and/or the activities of AspAP and GluAP progressively increase with the degree of saturation of fatty acids in the diet [4,7,10,72]. GluAP activity is modified by the composition of fatty acids in the diet and by the cholesterol content [4,5,6,9,57], directly or indirectly, and with an important role in the development of cardiovascular disorders and the pathophysiology of hypertension. An analysis of grouped data showed significant correlations of angiotensinase activities in various cardiovascular tissues and plasma (Appendix A); however, a stratified analysis isolating dietary groups showed that angiotensinase activities were not associated with each group of diets and that they were not associated with the type of fat used. This absence of correlations does not agree with other studies in which they showed the impact of the fatty acid profile of the diet on central tissues and its correlation with APs activities [26,77]. It is demonstrated that the profile of fatty acids and the levels of APs activities are modified depending on the type of fat used [77], and their activities correlate with dietary fat composition [7,26]. 

The cardiac interstitium is made up of non-myocytic cells embedded in a highly organized extracellular matrix that contains a three-dimensional collagen network that serves to maintain the architecture of the myocardium and determine its rigidity [78]. It is known that circulating and myocardial cells RAS are directly involved in the regulation of cardiac interstitial remodeling. Oleic acid (n-9), a component of olive oil, prevents Ang II-induced cardiac remodeling (fibrosis and hypertrophy) by suppressing the expression of collagen and fibroblast growth factor 23 (FGF23) in mice [79]. GluAP inhibitors and Ang IV analogs prevent cardiac dysfunction by normalizing central/local GluAP hyperactivity and attenuating cardiac hypertrophy and fibrosis [27,80,81]. Our results show that together with the marked decrease in the RAS activity of the VOO diet compared with the Bch diet, in addition, the VOO diet presented normal values of DPP-IV activity (mb) in the atrium and aorta, but they were significantly lower with the diet Bch. In summary, our data showed higher values of RAS regulatory activities together with lower values of DPP-IV with the Bch diet; these data coincide with investigations that reveal that Ang II stimulates collagen synthesis and inhibits collagenase activity in cardiac fibroblasts [78,82,83]. In a contradictory way, other researchers confirm that HFD with a high content of SAFA increases the expression and activity of DPP-IV in the aorta and atrium, playing a role in the development of aortic stiffness, vascular oxidative stress, endothelial dysfunction, and vascular remodeling by promoting increased deposition of collagen fibers [84,85,86]. The grouped correlation between DPP-IV and angiotensinases activities in cardiovascular tissues (Table 1) manifested a strong association with collagenase DPP-IV activity when the classical and non-classical pathways of RAS were activated. A stratified analysis between independent groups confirmed the association of activities with the variable diet. These results are interesting because it was determined that olive oil has an important role in the association of regulatory activities of RAS (AlaAP, ArgAP, AspAP, CysAP, GluAP) and DPP-IV activity in the atrium and ventricle (Table 1). Second, as we have commented previously, the VOO diet detected in the atrium the lowest values of CysAP activity and normal values of DPP-IV activity compared with the Bch diet. However, the atrium showed a marked increase in CysAP activity with diet Bch compared with diet S and lower values of DPP-IV activity with diet Bch versus diet VOO. Without going further, strong correlations of grouped data were detected between DPP-IV and CysAP activities in all tissues (Table 1), where a stratified analysis showed that the VOO diet was a variable that affected this association in the atrium and ventricle. These results make sense since it is known that there is an interaction of CysAP with DPP-IV that acts in the pathway of the kinin system (metabolism of bradykinin) as a counter-regulation of an overactivation of the RAS [87]. In non-cardiovascular tissues with fibrosis, they are also represented by inflammation and a deposit of cellular matrix, where DPP-IV has a profibrotic role of binding to fibronectin, influencing cell adhesion and migration [88,89]. 

With the VOO diet vs. the Bch diet, in the ventricular and aortic membrane-bound fractions, the lower values of GluAP and ArgAP angiotensinase activities allowed less arterial damage, as reflected by the activity of the functional marker LeuAP. The LeuAP (mb) activity was elevated in aorta with the Bch diet compared with the S diet; however, the VOO diet showed its benefit by maintaining the LeuAP activity at normal values. Similar results were found in another vascularized organ model using the same type of experimental diets [9]. Our experimental diets did not show significant GGT activity in the cardiovascular tissue fractions; therefore, it does not indicate increased oxidative stress or insufficient oxidation. These results are totally contradictory to the values found in different vascularized organs of animals treated with the same experimental diets, where the VOO diet had lower values than the Bch diet, even behaving as its control [7,9]. 

The pGluAP (mb) activity was significantly reduced in the atrium with both HFDs with respect to diet S. These data do not agree with other results, where a significant increase in pGluAP (sol/mb) activity of the atrium has been described in renovascular hypertension; however, in the aorta, there are no differences, and in the ventricle, the activity is reduced without being significant with respect to its healthy controls [17]. These differences are probably due to the differences between the animal models. 

Treatment of hypertension with AP inhibitors [4,5] has not been shown to totally prevent CVD outcomes [90,91]. Angiotensinases and DPP-IV are objects of study to find new therapeutic approaches targeting RAS and associated peptides in hypertension and heart failure [87]. However, prevention with diets enriched with olive oil has an important influence on the reduction in the central and local RAS pathway [6,7,9,10]. Our results also showed lower angiotensinase activity values with the VOO diet compared with the Bch diet. A main limitation of our study is that the actions of HFDs can be different depending on many variables, such as the species and animal model used, as well as depending on the multiple components of the diet itself; therefore, these experiments cannot be transferred to humans. Adding to that accentuation of the effects of RAS, the experiments were started with adult animals (6 months of age), and the experimental diets were continued with 6-month follow-up. The older age among the animals used in the experiment and the small size of the samples (*n* = 6–5/group) can increase the variability in the parameters studied and contribute to reducing the differences observed between the groups. Additional research on components of olive oil should be conducted in new preclinical models associated with CDV. In addition, it should be determined in more detail if the cardiovascular benefits of olive oil derive from saponifiable lipids or non-saponifiable lipids, which are minor components, such as phenolic compounds with antioxidant character. For this, it is proposed to try a pure diet in MUFA (refined virgin olive oil) or extracts of olive leaves rich in polyphenols. 

## 5. Conclusions

All the results predicted a differential effect of two high-fat diets on RAS regulation, functionality, and stability in the atrium, ventricle, and aorta (Figure 5). The inclusion of virgin olive oil in the diets moderated the normalization of RAS activities in the atrium (CysAP), ventricle (GluAP and ArgAP), and aorta (GluAP and ArgAP). However, the Bch diet had a significant increase in CysAP activity in the atrium, together with increases in the activity of functional marker LeuAP in the ventricle and aorta, while the olive oil normalized the LeuAP values. The VOO diet showed normal DPP-IV activity; however, the Bch diet reduced DPP-IV activity in the aorta. Interestingly, DPP-IV and angiotensinase activities were strongly correlated intragroup, especially with the VOO diet for AlaAP, ArgAP, AspAP, CysAP, and GluAP activities in the atrium ventricle. The replacement of saturated fats with monounsaturated fats would benefit by normalizing the activities that regulate homeostasis of the classical and non-classical pathways of RAS and DPP-IV for proper maintenance and functioning of the cardiovascular system.

## Figures and Tables

**Figure 1 biomedicines-09-01149-f001:**
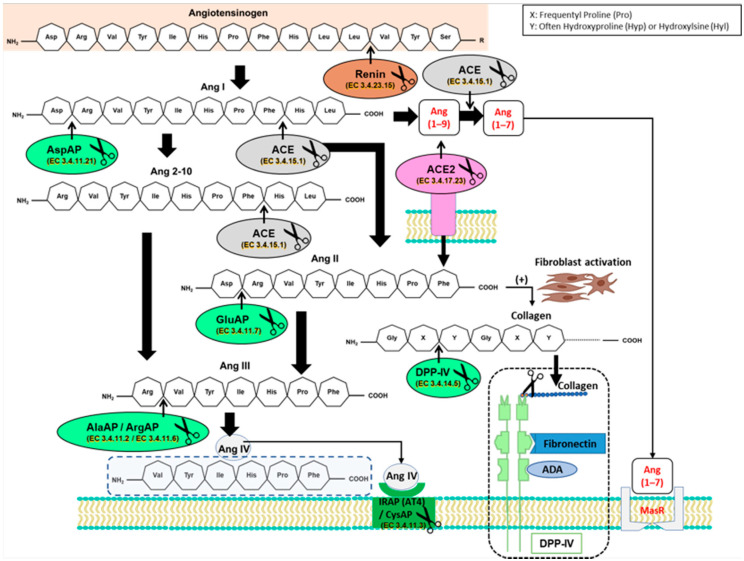
Partial diagram of angiotensinase activity downstream of the classical axis of the renin–angiotensin system (RAS) and the ACE2/Ang (1–7)/Mas receptor pathway, displaying the metabolic steps in which angiotensinase activities are involved. Ang II stimulates fibroblasts to produce collagen, this being a substrate for dipeptidyl peptidase IV (DPP-IV). ACE: angiotensin-converting enzyme; ACE2: angiotensin-converting enzyme 2; ADA: adenosine desaminase; AlaAP: alanyl aminopeptidase; Ang I: angiotensin I; Ang (1–7): angiotensin (1–7); Ang (1–9): angiotensin (1–9); Ang 2–10: angiotensin 2–10; Ang II: angiotensin II; Ang III: angiotensin III; Ang IV: angiotensin IV; ArgAP: arginyl aminopeptidase; AspAP: aspartyl aminopeptidase; AT4: angiotensin AT4 receptor; GluAP: glutamyl aminopeptidase; IRAP/CysAP: insulin-regulated aminopeptidase activity/cystinyl aminopeptidase; MasR: Mas receptor. The scissors symbol represents proteolytic activity of the enzyme, cleaving amino acids from the amino terminals.

**Figure 2 biomedicines-09-01149-f002:**
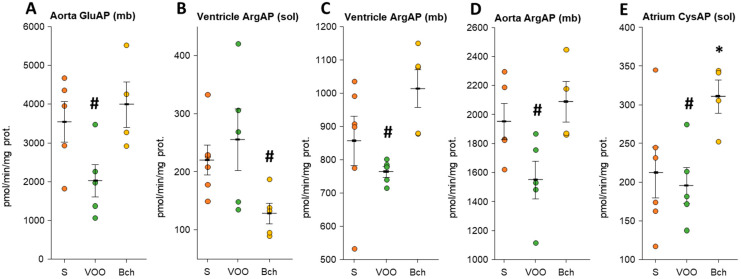
Mean values ± standard errors of significant (**A**) glutamyl aminopeptidase (GluAP), (**B**–**D**) arginyl aminopeptidase (ArgAP), (**E**) cystinyl aminopeptidase (CysAP) activities in soluble (sol) and membrane-bound (mb) fractions of atrium, ventricle, and aorta. The small colored dots are data points per group. Values are expressed as pmol/min/mg prot in the rest of the tissues analyzed. * indicates significant differences between virgin olive oil diet (VOO) or butter plus cholesterol diet (Bch) vs. standard diet (S). * *p* < 0.05. # indicates significant differences between VOO and Bch, # *p* < 0.05.

**Figure 3 biomedicines-09-01149-f003:**
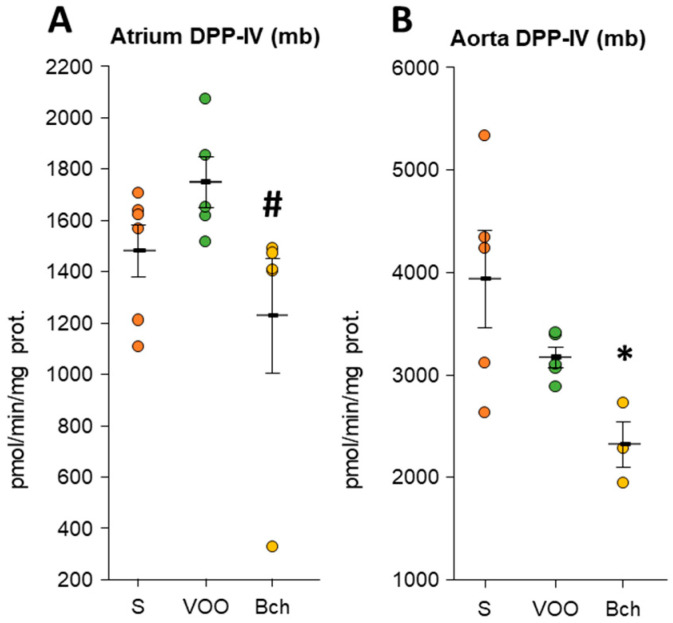
Mean values ± standard errors of significant (**A**,**B**) dipeptidyl peptidase IV (DPP-IV) activity in membrane-bound (mb) fraction of atrium and aorta. The small colored dots are data points per group. Values are expressed as pmol/min/mg prot in the rest of the tissues analyzed. * indicates significant differences between virgin olive oil diet (VOO) or butter plus cholesterol diet (Bch) vs. standard diet (S). * *p* < 0.05. # indicates significant differences between VOO and Bch, # *p* < 0.05.

**Figure 4 biomedicines-09-01149-f004:**
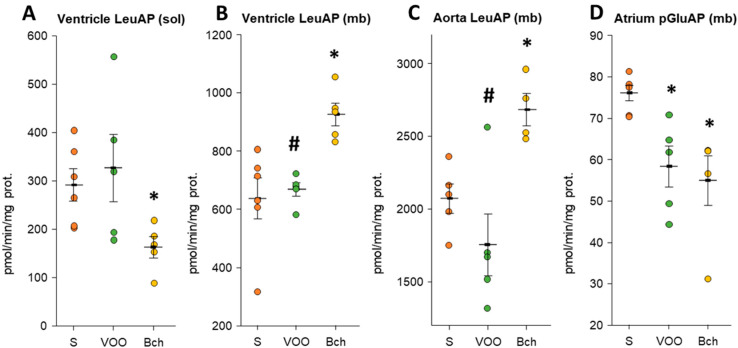
Mean values ± standard errors of significant (**A**–**C**) leucyl aminopeptidase (LeuAP) and (**D**) pyroglutamyl aminopeptidase (pGluAP) activities in soluble and membrane-bound (mb) fractions of atrium, ventricle, and aorta. The small colored dots are data points per group. Values are expressed as pmol/min/mg prot in the rest of the tissues analyzed. * indicates significant differences between virgin olive oil diet (VOO) or butter plus cholesterol diet (Bch) vs. standard diet (S), * *p* < 0.05. # indicates significant differences between VOO and Bch, # *p* < 0.05.

**Figure 5 biomedicines-09-01149-f005:**
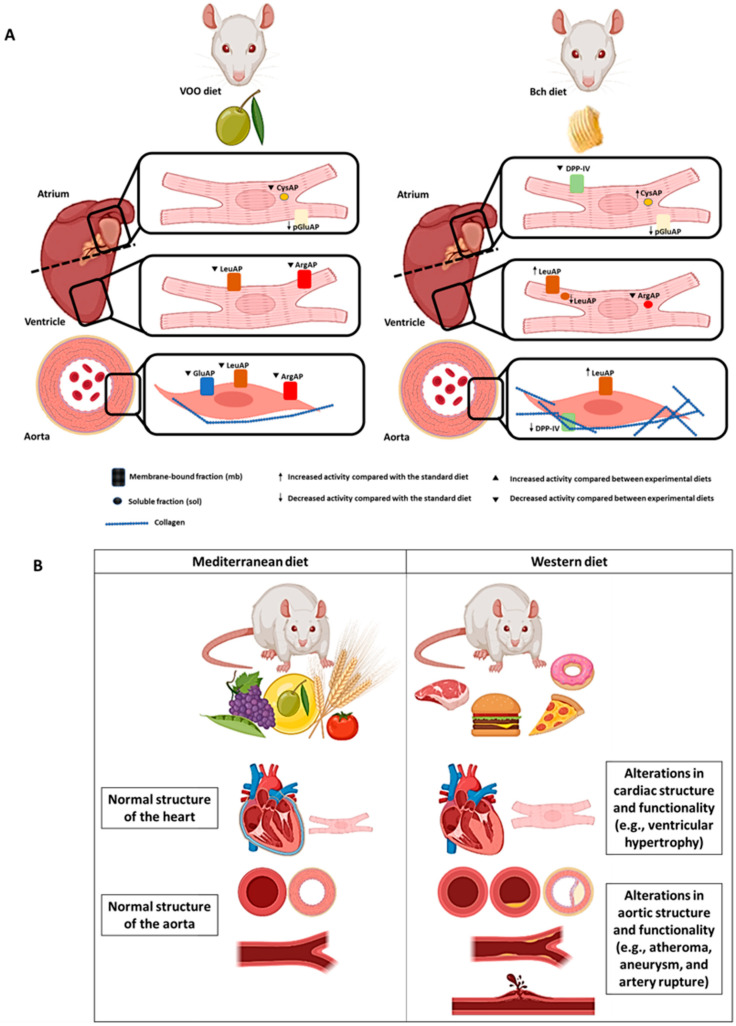
(**A**) Summary of the influence of the type of fat in the diet on the regulatory activities of the renin–angiotensin system, cardiac biomarker leucine aminopeptidase, and dipeptidyl peptidase IV. ArgAP: arginyl aminopeptidase; CysAP: cystinyl aminopeptidase; DPP-IV: dipeptidyl peptidase IV; GluAP: glutamyl aminopeptidase; LeuAP: leucyl aminopeptidase; pGluAP: pyroglutamyl aminopeptidase. VOO: diet enriched with virgin olive oil; Bch: diet enriched with butter plus cholesterol. (**B**) Differences in cardiovascular physiology with a long-term Mediterranean diet or Western diet at the structural or functional level in the heart and aorta.

**Table 1 biomedicines-09-01149-t001:** Significant correlations of dipeptidyl peptidase IV (DPP-IV) activity with angiotensinase activities in atrium, ventricle, and aorta.

Sample	DPP-IV (Fraction)	Angiotensinase (Fraction)	*p*-Value	R	Intra-Group Correlations	*p*-Value	R
Atrium	DPP-IV (sol) vs.	AlaAP (sol)	0.004	0.692	VOO–VOO	0.007	0.969
		ArgAP (sol)	0.002	0.734			
		CysAP (sol)	<0.001	0.777	S–S VOO–VOO	0.008 0.023	0.927 0.928
Atrium	DPP-IV (mb) vs.	AlaAP (mb)	0.012	0.609	S–S Bch-Bch	0.014 0.043	0.902 0.890
		ArgAP (mb)	0.012	0.609	S–S	0.011	0.912
		CysAP (mb)	0.040	0.517			
		GluAP (mb)	0.037	0.526	S-S	0.006	0.936
Ventricle	DPP-IV (sol) vs.	AlaAP (sol)	<0.001	0.942	VOO–VOO Bch–Bch	0.002 0.018	0.984 0.940
		ArgAP (sol)	<0.001	0.923	VOO–VOO	0.004	0.979
		AspAP (sol)	<0.001	0.813	S–S VOO–VOO	0.040 0.046	0.833 0.954
		CysAP (sol)	<0.001	0.756	VOO–VOO	0.023	0.929
		GluAP (sol)	<0.001	0.904	VOO–VOO	0.002	0.986
Aorta	DPP-IV (sol) vs.	ArgAP (sol)	0.048	0.542			
		AlaAP (sol)	0.044	0.589			
Aorta	DPP-IV (mb) vs.	CysAP (mb)	0.019	0.638			

Note: The values represent only those significant correlations between tissues. Values represent correlation of dipeptidyl peptidase IV vs. angiotensinases activities, in soluble (sol) and membrane-bound (mb) fractions of atrium, ventricle, and aorta. Intra-group correlations analyze the correlations between activities of the same dietary group. AlaAP: alanyl aminopeptidase; ArgAP: arginyl aminopeptidase; AspAP: aspartyl aminopeptidase; CysAP: cistinyl aminopeptidase; GluAP: glutamyl aminopeptidase; DPP-IV: dipeptidyl peptidase IV. *p*-value less than 0.05 was considered significant. R: linear correlation coefficient.

## Data Availability

Not applicable.

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
