# Peer review of "High-Fat Diets Modify the Proteolytic Activities of Dipeptidyl-Peptidase IV and the Regulatory Enzymes of the Renin–Angiotensin System in Cardiovascular Tissues of Adult Wistar Rats"

_biomedicines, 2021, doi:10.3390/biomedicines9091149_

Round 1
Reviewer 1 Report
This report entitled 'High-fat diets modify the proteolytic activities of dipeptidyl-peptidase IV and the regulatory enzymes of the renin-angiotensin system in cardiovascular tissues of adult Wistar rats' suggested interesting topics in medicine and showed the role of two sources of dietary fat on the aminopeptidase activities involved in the regulation of the RAS.
Line 177
The reviewer thought that it was more adequate to set standard diet isocaloric compared to HFD diets for exact evaluation of the effects of fat as control. Please specify the reason not to set isocaloric, but identical weight.
Also, considering the description of line 165, it sounds like that there is no control group in this study. Please revise the contents appropriately for better experimental design.
Author Response
Authors’ Response to the Reviewer’s comments
Journal: International Journal of Molecular Sciences
Title of Paper: High-fat diets modify the proteolytic activities of dipeptidyl-peptidase IV and the regulatory enzymes of the renin-angiotensin system in cardiovascular tissues of adult Wistar rats
Authors: Germán Domínguez-Vías, Ana Belén Segarra, Manuel Ramírez-Sánchez, Isabel Prieto
Date Sent: 17 August, 2021
Dear Reviewer,
We appreciate your time and efforts in reviewing our manuscript. We are delighted to answer your question.
Yours faithfully,
Germán Domínguez-Vías
Reviewer 1:
Comments and Suggestions for Authors
This report entitled 'High-fat diets modify the proteolytic activities of dipeptidyl-peptidase IV and the regulatory enzymes of the renin-angiotensin system in cardiovascular tissues of adult Wistar rats' suggested interesting topics in medicine and showed the role of two sources of dietary fat on the aminopeptidase activities involved in the regulation of the RAS.
Dear reviewer, thank you very much for your evaluation.
Line 177
The reviewer thought that it was more adequate to set standard diet isocaloric compared to HFD diets for exact evaluation of the effects of fat as control. Please specify the reason not to set isocaloric, but identical weight. Also, considering the description of line 165, it sounds like that there is no control group in this study. Please revise the contents appropriately for better experimental design.
We appreciate your valuable comments. The aim of the work was to study the effect of high-fat diets (HFD) for that HFD was compared with a standard with the normal amount of fat supplied to laboratory rodents. And secondly, to compare different HFDs but with different fat sources, the standard diet was enriched with olive oil, as an example of the Mediterranean diet (high in monounsaturated fatty acids) and on the other hand the standard diet enriched with butter as an example of Western diet (high in saturated fatty acids and cholesterol).
Reviewer 2 Report
- The abstract is long and confusing and did not adhere to MDPI guidelines, max 200 word, and to read as one paragraph
- The introduction section is broad, should be rewritten not to exceed 1,000 word, in a way to be focused on the topic, and include clearly the aim of the study in addition to a potential hypothesis.
- The presentation of results needs to be cut and simplify represented.
- The discussion section should be rewritten, since now appears to be speculative. The needed structure is as follows:
- The main findings of the study, and their comparison with previous published literature
- The implication of these findings, and their translation in humans. Keeping in mind (and this should be clearly mentioned), that only 10% of experiments conducted in animals will be implemented in humans.
- The strengths and limitations of the study
- The future directions for future researches
Author Response
Authors’ Response to the Reviewer’s comments
Journal: International Journal of Molecular Sciences
Title of Paper: High-fat diets modify the proteolytic activities of dipeptidyl-peptidase IV and the regulatory enzymes of the renin-angiotensin system in cardiovascular tissues of adult Wistar rats
Authors: Germán Domínguez-Vías, Ana Belén Segarra, Manuel Ramírez-Sánchez, Isabel Prieto
Date Sent: 17 August, 2021
Dear Reviewer,
We appreciate your time and efforts in reviewing our manuscript. All the issues indicated in the comments from Reviewer have been addressed.
We have reduced and simplified the content of the manuscript. Reviewer 'suggestions, are marked in yellow in the text. Also at the request of the reviewers, we have indicated only the figures that are significant, and the totality of the data can be consulted as supplementary material. Major changes, made according to the Reviewer 'suggestions, are marked in yellow in the text.
Yours faithfully,
Germán Domínguez-Vías
Reviewer 2:
Comments and Suggestions for Authors
The abstract is long and confusing and did not adhere to MDPI guidelines, max 200 word, and to read as one paragraph
We thank the reviewer for the warning of the abstract format. The abstract has been rewritten reducing its size to one paragraph.
The introduction section is broad, should be rewritten not to exceed 1,000 word, in a way to be focused on the topic, and include clearly the aim of the study in addition to a potential hypothesis.
We apologize for the length of the introduction. A new version of the introduction has been rewritten, reducing the content and highlighting the importance of our subject of study. We have included a paragraph with the objectives and hypotheses (lines 95-104).
The presentation of results needs to be cut and simplify represented.
We have simplified the results indicating only the graphs with significant results and their values in the text, where all the angiotensinase activities are collected in a single section. All data (both significant and not) from all activities, as well as non-significant intragroup correlations, have been added as supplementary material in tables and figures.
The discussion section should be rewritten, since now appears to be speculative. The needed structure is as follows:
The main findings of the study, and their comparison with previous published literature
Thank you for your valuable comment. We have completely rewritten the discussion adjusting to his recommendation. We have stuck to comparing our results with what has been published and we have eliminated all speculation.
The implication of these findings, and their translation in humans. Keeping in mind (and this should be clearly mentioned), that only 10% of experiments conducted in animals will be implemented in humans.
We have added a paragraph to explain the difficulty of still transferring these experiments to humans given all the variables found. The objectives of this work continue to be encompassed in basic research to better understand the mechanisms underlying the regulation of the renin-angiotensin system in the face of different types of saturated fatty acids in the diet, as well as their implication on DPP-IV activity and its association with angiotensins linked to dietary groups.
Paragraph (lines 367-374):
“A main limitation of our study is that the actions of HFDs can be different depending on many variables, such as the species and animal model used, and the multiple components of the diet itself, therefore these experiments cannot be transferred to humans. Add that to accentuate the effects of RAS, experiments were started with adult animals (6 months of age) and the experimental diets were continued with 6-month follow-up. The high age among the animals used in the experiment and the small size of the samples (n = 6-5/group) can increase the variability in the parameters studied and contribute to reducing the differences observed between the groups”.
The strengths and limitations of the study:
The future directions for future researches
The authors have added to the discussion a future approach that we would like to propose to shed light on many of the questions of this work in models associated with cardiovascular diseases using only MUFA (refined saponifiable lipid from olive oil) or non-saponifiable components (phenolic compounds of olive leaf extracts), to better understand the regulatory mechanism of the classical and non-classical pathways of the renin-angiotensin system (RAS) and its direct relationship with DPP-IV as a countermeasure (or not) to the activation of RAS.
Paragraph (lines 374-379):
“Additional research on components of olive oil should be conducted in new preclinical models associated with CDV. In addition, it should be deepened in more detail if the cardiovascular benefits of olive oil derive from saponifiable fraction or non-saponifiable fraction, which are minor components such as phenolic compounds with antioxidant character. For this, it is proposed to try a pure diet in MUFA (refined virgin olive oil) or extracts of olive leaves rich in polyphenols”.
Reviewer 3 Report
In this study, Domínguez-Vías et al. investigated the effects of high-fat diets on the enzyme activities of dipeptidyl peptidase IV in the regulation of the renin-angiotensin system. While the authors’ findings may be potentially of interest, the results were mainly based on correlation, and most data showed no significance in three groups (one standard chow, two high-fat diets). This raises major concerns in drawing conclusions in that mere correlation may not result from direct effects nor from causal mechanisms. In addition, the main significant finding of the effects of high-fat diets in this study was somehow only LeuAP (Fig. 2) and GluAP (Fig. 3). Validation on these findings were not provided. Thus, the importance of overall findings lacks significant new insights. At least, the authors should examine and compare the lipid profiles of three groups of rats. There were same errors in data analysis of Figs 4, 6, 8, 10 and 12. The data points from three groups should be labeled and analyzed independently, not pooled together, for group comparisons. In the abstract, the authors would like to link their findings to the benefits from taking Mediterranean diet, but the authors did not provide any supporting evidence to make conclusions. At least, a group of rats fed with MUFA should be included in the experiments for comparison with other groups.
Other suggestions:
-Figs. 3, 5, 7, 9, 11, 13, 15 and 16. All data points in each group should be displayed in the bar graphs. The actual p-values between compared groups in A to D should be shown. They can also be illustrated in a table. If marginal differences were observed, and yet statistical significance were seen, the authors might want to increase animal numbers in groups, accordingly.
-Figs. 4. As mentioned above, data of three diet groups should be labeled with different symbols for analysis in terms of the glutamyl aminopeptidase (GluAP) activities between aorta and ventricle (soluble vs. membrane), otherwise the results were misleading. A total of 16 rats (6, 5, 5) were listed, but only 14 samples were analyzed and shown in the graph.
-Fig. 6. The alanyl aminopeptidase (AlaAP) activities of the soluble fraction in ventricle and plasma of three experimental groups should be labeled and analyzed independently.
- Similar comments to Figs 4 and 6 are given to Figs.8, 10, 12 and 14 in linear regression analysis.
Author Response
Authors’ Response to the Reviewer’s comments
Journal: International Journal of Molecular Sciences
Title of Paper: High-fat diets modify the proteolytic activities of dipeptidyl-peptidase IV and the regulatory enzymes of the renin-angiotensin system in cardiovascular tissues of adult Wistar rats
Authors: Germán Domínguez-Vías, Ana Belén Segarra, Manuel Ramírez-Sánchez, Isabel Prieto
Date Sent: 17 August, 2021
Dear Reviewer,
We appreciate your time and efforts in reviewing our manuscript. All the issues indicated in the comments from Reviewers have been addressed. We have modified the figures and added new results by performing intragroup stratified (partial) correlations. Major changes, made according to the Reviewers’ suggestions, are marked in yellow in the text.
Yours faithfully,
Germán Domínguez-Vías
Reviewer 3:
Comments and Suggestions for Authors
In this study, Domínguez-Vías et al. investigated the effects of high-fat diets on the enzyme activities of dipeptidyl peptidase IV in the regulation of the renin-angiotensin system. While the authors’ findings may be potentially of interest, the results were mainly based on correlation, and most data showed no significance in three groups (one standard chow, two high-fat diets). This raises major concerns in drawing conclusions in that mere correlation may not result from direct effects nor from causal mechanisms.
We appreciate the comment and make amends for our mistake. We agree with the reviewer that a high value of the correlation coefficient does not indicate causality. New stratified correlation analyzes have confirmed that there is no correlation among angiotensinase activities into a single diet. We have added this new correction (lines 204-208) indicating the non-implication of diet in the correlations between angiotensinases in the results section and as supplementary material (Supplementary Figure 1).
In addition, the main significant finding of the effects of high-fat diets in this study was somehow only LeuAP (Fig. 2) and GluAP (Fig. 3).
To improve the significance of the experimental diets on the results, we have indicated only the figures of the representative activities.
Validation on these findings were not provided. Thus, the importance of overall findings lacks significant new insights. At least, the authors should examine and compare the lipid profiles of three groups of rats.
I thank the reviewer for the importance of this question. Previous results that we have recently published verified the importance of supplementing the diet with olive oil (20%), obtaining multiple benefits on the lipid profile, oxidative stress and blood pressure (among others). In our laboratory using the same HFDs (VOO and Bch), we confirmed the beneficial effects in animals fed the VOO diet versus the pernicious results with the Bch diet. With the VOO diet, they do not increase their body weight, they do not increase the values of systolic blood pressure (SBP), they maintain normal values of triglycerides and the fraction of VLDL cholesterol, reduce total cholesterol and its LDL fraction, reduce oxidative stress and lipid peroxidation against the pernicious effects that the Bch diet does manifest [4,7,9].
We have added a comment about it in the text (lines 272-284).
There were same errors in data analysis of Figs 4, 6, 8, 10 and 12. The data points from three groups should be labeled and analyzed independently, not pooled together, for group comparisons.
We value your comment and share your opinion. The global results do not allow to observe the true structure of the data and lead to false conclusions, the angiotensinase-diet relationship in the population of interest is more complex. To avoid this type of problem, a stratification analysis was used where the analysis was divided independently into interest groups (according to the type of diet administered). The partial correlation was used to assess whether we are facing a spurious relationship due to the presence of a third confounding variable. Performing a stratified analysis eliminated the effect of diet in the event of a loss of association. This means that the association between the variables does not depend on the diet. We have added this new result in a summarized way in the text at the end of each subsection. We have detailed all this information as supplementary material (Supplementary Figures 1-3).
In the abstract, the authors would like to link their findings to the benefits from taking Mediterranean diet, but the authors did not provide any supporting evidence to make conclusions. At least, a group of rats fed with MUFA should be included in the experiments for comparison with other groups.
Olive oil was chosen as the main source of the MUFA of the Mediterranean diet. We have not used refined olive oil, whose composition would be pure MUFA.
Other suggestions:
-Figs. 3, 5, 7, 9, 11, 13, 15 and 16. All data points in each group should be displayed in the bar graphs. The actual p-values between compared groups in A to D should be shown. They can also be illustrated in a table. If marginal differences were observed, and yet statistical significance were seen, the authors might want to increase animal numbers in groups, accordingly.
We have reduced the figures to indicate only the significant ones, and we have improved the design of the figures as suggested by the reviewer, indicating all the data points in each group of diets for each enzyme activity. To consult the values of all the activities analyzed in each section, we have created supplementary tables together with their p-values (Supplementary Tables 1-3).
-Figs. 4. As mentioned above, data of three diet groups should be labeled with different symbols for analysis in terms of the glutamyl aminopeptidase (GluAP) activities between aorta and ventricle (soluble vs. membrane), otherwise the results were misleading. A total of 16 rats (6, 5, 5) were listed, but only 14 samples were analyzed and shown in the graph.
For simplicity, we have added under this type of correlation graph an intragroup table (association in each dietary group the activities of the tissues). As we have commented previously, the partial correlations (stratified) did not show that the separate diets had a direct implication between these associations. The total number of animals is indeed that. Certain graphs present overlapping points, giving the appearance of being a single point. It is also true that other graphs like the one you mentioned (Figure 4) have 14 points for presenting outliers or not detected.
-Fig. 6. The alanyl aminopeptidase (AlaAP) activities of the soluble fraction in ventricle and plasma of three experimental groups should be labeled and analyzed independently.
- Similar comments to Figs 4 and 6 are given to Figs.8, 10, 12 and 14 in linear regression analysis.
Done. I repeat myself to the previous points. An in-depth analysis of each graph (stratified correlation between dietary groups) showed that diet is not really involved in the association of these activities. These new arguments are included in the results at the end of each section.
Following the reviewer's recommendations, the authors have also independently decided to perform the same independent analyses between dietary groups (stratified correlations) with one of the most interesting results of the work: the association of DPP-IV activity with angiotensinase activities. These analyses reinforced the results, determining strong correlations between DPP-IV and angiotensinase activities, especially with the olive diet in the atrium and ventricle (Table 1).
Round 2
Reviewer 2 Report
Can be accepted in the current form.